# BioBlobs: Differentiable Graph Partitioning for Protein Representation Learning

## Abstract

Protein function is driven by coherent substructures which vary in size and topology, yet current protein representation learning models (PRL) distort these signals by relying on rigid substructures such as $k$-hop and fixed radius neighbourhoods. We introduce BioBlobs, a plug-and-play, fully differentiable module that represents proteins by dynamically partitioning structures into flexibly-sized, non-overlapping substructures ("blobs"). The resulting blobs are quantized into a shared and interpretable codebook, yielding a discrete vocabulary of function-relevant protein substructures used to compute protein embeddings. We show that BioBlobs's representations improve the performance of widely used protein encoders such as GVP-GNN across various PRL tasks. Our approach highlights the value of architectures that directly capture function-relevant protein substructures, enabling both improved predictive performance and mechanistic insight into protein function. Source code is available at `https://anonymous.4open.science/r/BioBlobs-EECD/`

## 1 Introduction

Proteins are structured macromolecules that drive fundamental biological processes across the tree of life. A protein's ability to carry out its function is rooted in its specific three-dimensional arrangement—its fold. Despite decades of painstaking curation and experimentation revealing many ways in which folds mediate biological processes, the vast majority of proteins in major databases remain without reliable functional annotation (Kustatscher et al., 2022).

For a long time, a key obstacle to assigning function was the difficulty of determining the structure for a given sequence. This bottleneck has been alleviated by highly accurate structure predictors such as the AlphaFold Jumper et al. (2021) and ESM (Lin et al., 2023) families, yielding databases with hundreds of millions of predicted structures. With structures now broadly available, the central challenge is to encode structural information at this unprecedented scale.

Protein representation learning (PRL) (Wu et al., 2022) addresses this by training neural models to capture salient structural and biophysical properties in compact embeddings that can inform downstream tasks. Given a protein, PRL methods typically proceed in two stages: (1) choose a basic "atom" or unit (e.g., a sequence span, a residue with its nearest neighbours, a voxel, or a secondary-structure element); and (2) encode each unit in the context of the whole to produce a final representation. These representations feed decoders for supervised tasks (e.g., binding-site detection, pathway membership, protein–protein interaction prediction, protein design) Gligorijević et al. (2021); Derry et al. (2025); Watson et al. (2023); Dauparas et al. (2022), or are learned via self-supervision to produce general-purpose encodings Zhang et al. (2022); Chen et al. (2024).

Existing PRL systems often operate over rigidly defined units (Fasoulis et al., 2021), yet protein function is driven by cohesive substructures that operate modularly (Rorick & Wagner, 2011). Among many examples are catalytic triads (Ser–His–Asp), Rossmann-like nucleotide-binding cores, and P-loop (Walker A/B) NTPase sites (Dodson & Wlodawer, 1998; Rossmann et al., 1974; Leipe et al., 2003). When the unit is fixed *a priori*—e.g., $k$-hop neighbourhoods or voxel grids—models can fracture functional assemblies and distort the learned representations. This would be like trying to understanding the mechanisms of a bicycle by cutting it into uniformly sized cubes. Likewise in proteins, functional modules rarely adopt such regular substructures. Compounding this, the appropriate units are context-dependent and vary in size and topology. Moreover, the atomization step

is inherently discrete (e.g., selecting a residue and its neighbours), making it non-differentiable and therefore typically excluded from end-to-end training. The atoms we seek should be (1) connected substructures of the protein, (2) contributory to function, and (3) distributed modularly, reflecting the conservative forces of evolution.

We cast the problem of atomising a protein into cohesive substructures as a graph-partitioning task (Buluç et al., 2016). Protein graphs—capturing neighbourhood relations and spatial organisation among residues or atoms—are standard in PRL, with GVP-GNNs being a widely used example (Jing et al., 2021). Graph partitioning assigns each node to exactly one part, producing a disjoint cover of the graph; on residue/atom graphs, such partitions yield non-overlapping groups—precisely the structural units we aim to model. Because the space of partitions grows exponentially with the number of nodes, many natural formulations are NP-hard (Feder et al., 1999). Nonetheless, recent advances in neural combinatorial optimisation over graphs learn probability distributions on nodes/edges to construct powerful probabilistic algorithms for tasks such as set cover, alignment, and compression (Karalias & Loukas, 2020; Bouritsas et al., 2021). These ideas remain largely unexplored within PRL.

Here we propose BIOBLOBS, which performs principled partitioning of 3D protein graphs into cohesive substructures.

- **Neural partitioning layer for PRL.** We introduce BIOBLOBS, a plug-and-play neural graph-partitioning layer that represents proteins as learnable functions of their **coherent substructure units**, guarantees connectivity, adapts granularity to the task, and can be placed atop established encoders (e.g., standard GNNs and GVP-based models). Parameter sharing enables the model to distil recurring substructures, reflecting the natural modularity of proteins.

- **Strong empirical performance.** BIOBLOBS achieves top performance on three established protein-function benchmarks under rigorous controls for structure leakage, outperforming strong PRL baselines.

- **Ablations on granularity.** We systematically vary the maximum substructure size and the number of partitions, providing insight into organisational principles of protein structure and function.

- **Interpretability.** Importance scores over discovered substructures enable higher-order interpretability, linking predictions to human-understandable functional units.

## 2 BIOBLOBS

The BIOBLOBS layer dynamically computes a hard assignment of residues to distinct substructures (sets of residues) which we term "blobs". We then compute representations of each blob and incorporate global context to arrive at a final embedding for the whole protein.

### 2.1 PROTEIN GRAPH CONSTRUCTION

We represent each protein structure as a geometric graph $\mathcal{G} = (\mathcal{V}, \mathcal{E})$, where nodes $v_i \in \mathcal{V}$ correspond to amino acid residues and edges $(i, j) \in \mathcal{E}$ connect residues that are spatially proximate, determined by the $k$-nearest neighbors of the $C_\alpha$ coordinates. Each node $v_i$ is associated with geometric features $\mathbf{h}_i = (\mathbf{s}_i, \mathbf{V}_i)$, derived from the coordinates of the $O, C, C_\alpha, N$ atoms in the residue. These features consist of scalar components $\mathbf{s}_i \in \mathbb{R}^{d_s}$ (e.g., dihedral angles) and vector components $\mathbf{V}_i \in \mathbb{R}^{d_v \times 3}$ (e.g., orientations). Similarly, each edge is assigned features $\mathbf{h}_{ij}^E = (\mathbf{s}_{ij}, \mathbf{V}_{ij})$, where $\mathbf{s}_{ij} \in \mathbb{R}^{d_e}$ denotes scalar values such as inter-residue distances, and $\mathbf{V}_{ij} \in \mathbb{R}^{d_{ev} \times 3}$ represents vector quantities such as spatial directions.

### 2.2 PROTEIN STRUCTURE ENCODER

We employ Geometric Vector Perceptron (GVP) layers (Jing et al., 2021) to obtain initial residue-level features while preserving geometric equivariance. The structure encoder consists of $L$ GVP convolution layers:

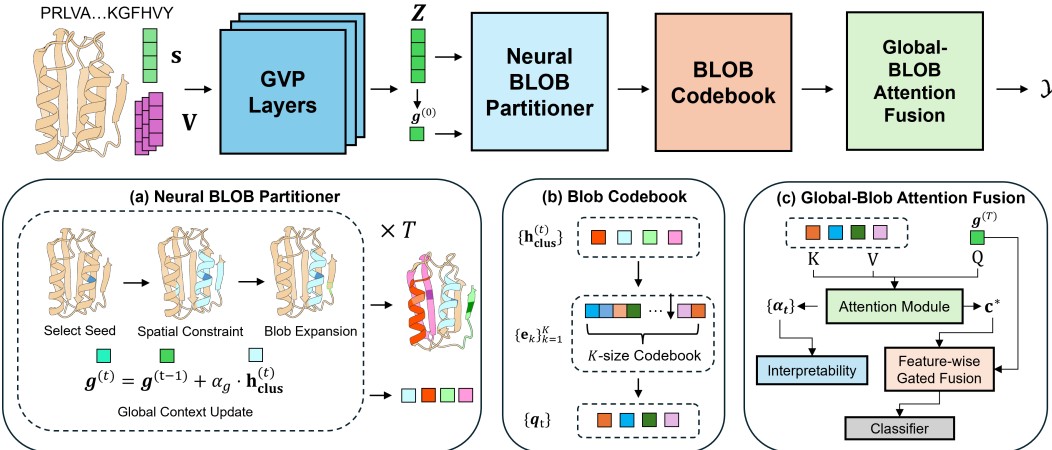

Figure 1: Overview of the BIOBLOBS pipeline. The framework consists of four main components: a protein encoder, a neural partitioner, a blob codebook, and a global-blob attention fusion module. The GVP encoder first processes the protein graph and produces residue embeddings. **(a) Neural Blob Partitioner**. A seed residue is first selected with Gumbel–Softmax. Its $k$-hop neighborhood is then identified to restrict the candidate pool. Finally, a blob expander scores the candidates and assigns residues to form cohesive local substructures. **(b) Blob Codebook**. The resulting blob embeddings are quantized into a discrete codebook that captures frequent and label-relevant protein substructures. **(c) Global–Blob Attention Fusion**. The quantized blob embeddings are integrated with the global feature using a multi-key attention mechanism. This produces both a fused representation for classification and an interpretable importance score distribution over blobs.

$$\mathbf{h}_i^{(\ell+1)} = \text{GVPConv}\left(\mathbf{h}_i^{(\ell)}, \left\{\mathbf{h}_j^{(\ell)} : j \in \mathcal{N}(i)\right\}, \mathbf{h}_{ij}^E\right) \tag{1}$$

where $\mathcal{N}(i)$ denotes the neighbors of node $i$. After $L$ layers, we obtain updated scalar node features through a projection layer, $\mathbf{z}_i = \text{Proj}\left(\mathbf{s}_i^{(L)}\right) \in \mathbb{R}^D$, which are then used to guide the partition module.

## 2.3 NEURAL BLOB PARTITIONER

Given residue embeddings $\mathbf{Z} = [\mathbf{z}_1, \cdots, \mathbf{z}_N] \in \mathbb{R}^{N \times D}$ for a protein with $N$ residues, we first derive an initial global context vector $\mathbf{g}^{(0)}$ by pooling all residue embeddings. The partitioner then constructs up to $T$ blobs through an iterative differentiable process. Initially, we create an assignment matrix $\mathbf{M}^{(0)} \in \{0, 1\}^{N \times T}$, which records the residue composition of each blob at iteration $t \in [1, 2, \cdots, T]$. An availability mask $\mathbf{m}^{(0)} \in \{0, 1\}^N$ is also maintained to keep track of unassigned residues and prevent overlapping assignments. At each iteration $t > 0$, we proceed discrete, differentiable partition (Bouritsas et al., 2021) with three key steps.

**Seed Residue Selection.** We begin by using a seed scoring network to predict logits over all residues: $\boldsymbol{\ell}^{(t)} = f_{\text{seed}}([\mathbf{Z}; \mathbf{g}^{(t-1)}]) \in \mathbb{R}^N$. Unavailable residues are masked by setting their logits to $-\infty$ using $\mathbf{m}^{(t-1)}$, ensuring that only unassigned residues remain candidates. We then sample from a relaxed categorical distribution using Gumbel–Softmax (Jang et al., 2016):

$$\mathbf{w}^{(t)} = \text{softmax}\left(\frac{\tilde{\ell}^{(t)} + \boldsymbol{\eta}}{\tau^{(t)}}\right), \qquad \eta_i \sim \text{Gumbel}(0, 1), \tag{2}$$

where the temperature $\tau^{(t)} = \max(\tau_{\min}, \tau_{\text{init}} \cdot \gamma^t)$ gradually decreases, shifting the sampling from exploratory to more deterministic behavior during training. To obtain a discrete seed while retaining gradients, we apply the straight-through (ST) estimator (Bengio et al., 2013):

$$\mathbf{e}^{(t)} = \text{one\_hot}\big(\arg\max_i w_i^{(t)}\big), \qquad \tilde{\mathbf{w}}^{(t)} = \mathbf{e}^{(t)} + \big(\mathbf{w}^{(t)} - \text{sg}(\mathbf{w}^{(t)})\big),$$

$$\mathbf{z}_{\text{seed}}^{\text{ST}} = \sum_{i=1}^{N} \tilde{w}_i^{(t)}\, \mathbf{z}_i. \tag{3}$$

In the forward pass, the hard one-hot vector $\mathbf{e}^{(t)}$ is used as the actual seed. In the backward pass, $\mathbf{w}^{(t)}$ is substituted so that gradients flow into the seed scorer and encoder. The straight-through seed embedding $\mathbf{z}_{\text{seed}}^{\text{ST}}$ thus acts as a continuous proxy for the discrete seed, enabling the partitioner to remain differentiable while still capturing the semantics of hard residue assignment.

**Threshold Prediction.** Given the seed residue $v_{\text{seed}}^{(t)}$, we derive its $k$-hop neighborhood and restrict to available residues $\mathcal{R}^{(t)} = \mathcal{N}_k\big(v_{\text{seed}}^{(t)}; \mathcal{E}\big) \cap \mathbf{m}^{(t-1)}$, which serves as the candidate set for expansion. A learned threshold head then predicts a per-blob threshold $\theta^{(t)}$ using the straight-through seed embedding $\mathbf{z}_{\text{seed}}$, the global context $\mathbf{g}^{(t-1)}$, and local structural statistics $\phi^{(t)}$ computed over $\mathcal{R}^{(t)}$:

$$\theta^{(t)} = f_{\text{thresh}}([\mathbf{z}_{\text{seed}}^{\text{ST}}; \mathbf{g}^{(t-1)}; \phi^{(t)}]), \tag{4}$$

This threshold adaptively controls how many candidates are included, thereby regulating the blob size in the subsequent expansion step.

**Blob Expansion.** Given the threshold $\theta^{(t)}$ and the candidate set $\mathcal{R}^{(t)}$, we expand around the seed by computing seed-conditioned scores, turning them into probabilities, forming straight-through hard memberships, and pooling a blob embedding that updates the global context.

$$s_i^{(t)} = \frac{\big(\mathbf{W}_k^{\text{Part}}\mathbf{z}_i\big) \cdot \big(\mathbf{W}_q^{\text{Part}}\mathbf{z}_{\text{seed}}^{\text{ST}}\big)}{\sqrt{d}}, \qquad v_i \in \mathcal{R}^{(t)}. \tag{5}$$

We map each score to an inclusion probability with a temperature-scaled sigmoid $p_i^{(t)}$, thresholded at 0.5 to obtain a binary membership $c_i^{(t)}$, and use a straight-through correction $\tilde{p}_i^{(t)}$ so gradients pass through the soft probability. To prevent a blob from growing unbounded, we further impose a maximum size $S$, keeping only the top-$S$ candidates ranked by their inclusion probabilities.

$$p_i^{(t)} = \sigma\big((s_i^{(t)} - \theta^{(t)})/\tau^{(t)}\big), \quad c_i^{(t)} = \mathbb{I}[p_i^{(t)} \geq 0.5], \quad \tilde{p}_i^{(t)} = c_i^{(t)} + (p_i^{(t)} - \text{sg}(p_i^{(t)})) \tag{6}$$

To obtain a differentiable blob representation while keeping discrete selection in the forward pass, we use a straight-through blob embedding that combines a hard mean and a soft mean as

$$\mathbf{h}_{\text{hard}}^{(t)} = \frac{\sum_i c_i^{(t)} \mathbf{z}_i}{\sum_i c_i^{(t)} + \varepsilon}, \quad \mathbf{h}_{\text{soft}}^{(t)} = \frac{\sum_i \tilde{p}_i^{(t)} \mathbf{z}_i}{\sum_i \tilde{p}_i^{(t)} + \varepsilon}, \quad \mathbf{h}_{\text{clus}}^{(t)} = \mathbf{h}_{\text{hard}}^{(t)} + \big(\mathbf{h}_{\text{soft}}^{(t)} - \text{sg}(\mathbf{h}_{\text{soft}}^{(t)})\big).$$

This straight-through form ensures that the forward computation uses the discrete blob for interpretability, while gradients flow through the soft weights to update the scorer and the encoder. The global context is updated in residual form, $\mathbf{g}^{(t+1)} = \mathbf{g}^{(t)} + \alpha_g f_{\text{global}}(\mathbf{h}_{\text{clus}}^{(t)})$, which accumulates information from discovered blobs without overwriting prior context and stabilizes training via the step size $\alpha_g$. Finally, $\mathbf{M}^{(t)}$ is updated according to the new blob partition. A concise step-by-step procedure is given in Algorithm 1.

## 2.4 Substructure Codebook

After the partitioner, we obtain blob embeddings $\mathbf{H} = [\mathbf{h}_{\text{clus}}^{(1)}, \ldots, \mathbf{h}_{\text{clus}}^{(T)}] \in \mathbb{R}^{T \times D}$. We introduce a vector–quantization codebook $\mathbf{E} = \{\mathbf{e}_k\}_{k=1}^{K} \subset \mathbb{R}^D$ to map each blob to a discrete substructure token (Van Den Oord et al., 2017; Razavi et al., 2019), steering the encoder and partitioner to discover

function–relevant substructures shared across the dataset. Let $\mathbf{h}_t$ denote the $t$-th blob embedding, the nearest–neighbor quantization selects its token as

$$k_t = \arg \min_{k \in \{1,\ldots,K\}} \|\mathbf{h}_t - \mathbf{e}_k\|_2^2, \qquad \mathbf{q}_t = \mathbf{e}_{k_t}. \tag{7}$$

We use the standard straight–through estimator so the forward pass uses the quantized vector while gradients update the encoder. To keep blob embeddings close to their selected codes, we use a commitment loss with stop–gradient from VQ-VAE-2 (Razavi et al., 2019). Besides, we add an entropy regularizer based on code usage probabilities $\{p_k\}_{k=1}^{K}$ to discourage dead codes and promote diverse usage:

$$\mathcal{L}_{\text{commit}} = \beta \sum_{t=1}^{T} \|\mathbf{h}_t - \text{sg}(\mathbf{q}_t)\|_2^2. \qquad \mathcal{L}_{\text{ent}} = \lambda_{\text{ent}} \sum_{k=1}^{K} p_k \log\left(\frac{p_k}{1/K}\right). \tag{8}$$

The code vectors are updated by an exponential moving average (EMA) during training, as shown in Appendix. B.3. The codebook outputs the quantized substructure representations form $\mathbf{H}_q = [\mathbf{q}_1, \ldots, \mathbf{q}_T] \in \mathbb{R}^{T \times D}$.

## 2.5 GLOBAL–BLOB ATTENTION FUSION

Finally, we integrate the quantized blob embeddings with the global protein representation using a single–query, multi–key attention mechanism (Vaswani et al., 2017), where $\mathbf{g} = \mathbf{g}^{(T)}$ serves as the query and $\mathbf{H}_q$ provides keys and values. We first project the query, keys, and values, and compute attention weights over blobs:

$$\mathbf{q} = \mathbf{W}_q\,\mathbf{g}, \quad \mathbf{K} = \mathbf{W}_k\,\mathbf{H}_q, \quad \mathbf{V} = \mathbf{W}_v\,\mathbf{H}_q, \quad \boldsymbol{\pi} = \text{softmax}\left(\frac{\mathbf{q}^\top \mathbf{K}}{\sqrt{D}}\right). \tag{9}$$

The weights $\boldsymbol{\pi} \in \mathbb{R}^T$ act as importance scores, indicating the relative contribution of each blob to the global representation. The weighted blob summary is then

$$\mathbf{c}^* = \mathbf{V}\,\boldsymbol{\pi} \in \mathbb{R}^D, \tag{10}$$

which allows the global context to attend to informative substructures. Finally, we fuse the global feature and the blob summary with a feature–wise gate:

$$\mathbf{h}_{\text{fuse}} = \text{LN}(\mathbf{g} + \boldsymbol{\beta} \odot \mathbf{W}_{\text{fuse}}\mathbf{c}^*), \qquad \boldsymbol{\beta} = \sigma(\text{MLP}([\mathbf{g};\mathbf{c}^*])), \tag{11}$$

where $\mathbf{W}_{\text{fuse}} \in \mathbb{R}^{D \times D}$ is a linear projection. The gate $\boldsymbol{\beta} \in (0, 1)^D$ adaptively balances the contributions of global features pooled from residue embeddings and local signals from the Blobs, and the fused representation $\mathbf{h}_{\text{fuse}}$ is passed to the classifier for prediction.

## 2.6 TIME COMPLEXITY ANALYSIS

The full pipeline after the protein encoder runs in near-linear time with respect to the number of residues. Among the components, the Neural Partitioner contributes the main cost, while the substructure codebook and global–blob attention add only modest overhead.

**Neural Partitioner.** Let $\mathcal{G} = (\mathcal{V}, \mathcal{E})$ denote the protein graph with $N = |\mathcal{V}|$ residues and $|\mathcal{E}| = O(k_{\text{NN}}N)$ edges, where $k_{\text{NN}}$ is the number of neighbors used in the $k$-nearest neighbor construction of the graph. The partitioner builds up to $T$ blobs through the iterative seed–threshold–expansion process, where $k_{\text{hop}}$ controls the radius of the neighborhood explored around each seed. At each iteration, three steps dominate the cost:

(i) *Seed selection:* the scorer computes logits over all $N$ residues and applies Gumbel–Softmax sampling, which is linear in the number of nodes, $O(N)$.

(ii) *Neighborhood construction:* the $k_{\text{hop}}$-hop neighborhood of the seed is obtained through a sparse traversal, which touches up to $k_{\text{hop}}$ layers of edges and costs $O(k_{\text{hop}}|\mathcal{E}|)$.

(iii) *Blob expansion:* each candidate in the neighborhood is scored by a seed-conditioned dot product, converted into probabilities, and thresholded, which again requires linear work in the number of candidates, bounded by $O(N)$.

Thus, the per-iteration complexity is $O(k_{\text{hop}}|\mathcal{E}| + N)$, and over $T$ iterations the total becomes

$$O\left(T\left(k_{\text{hop}}|\mathcal{E}| + N\right)\right) = O\left(T\left(k_{\text{hop}}\, k_{\text{NN}} + 1\right)N\right). \tag{12}$$

**Substructure Codebook.** For each of the $T$ blob embeddings with dimension $D$, the codebook computes squared distances to all $K$ code vectors. This requires $O(TKD)$ time, followed by nearest-neighbor assignment and an EMA update of the selected codes. Both operations share the same order of complexity, so quantization remains linear in the number of blobs and codes.

**Global–Blob Attention.** The fusion step integrates the global protein context with the $T$ blob embeddings. Projecting queries, keys, and values has cost $O((N + T)D^2)$, but since these projections are reused and $D$ is fixed, the dominant term is the attention operation itself. Computing attention scores across $T$ blobs with $H$ heads requires $O(THD)$, followed by weighted pooling and a feature-wise gating step in $O(D^2)$. Because $T \ll N$ and $H$ is fixed, the fusion module is lightweight compared to the partitioner.

## 2.7 MULTI-STAGE TRAINING

We adopt a two–stage training scheme to optimize BIOBLOBS in a stable and efficient manner.

- **Stage 1:** We first train the encoder, partitioner, and global–blob attention fusion while bypassing the substructure codebook. The original blob embeddings are used for fusion. This stage allows the model to form meaningful blobs relevant to protein function using only the cross–entropy loss.
- **Stage 2:** We initialize the codebook using $K$–means clustering on the blob embeddings learned in Stage 1, providing well–separated starting codes. The fusion module receives the quantized blob embeddings as input. The codebook is then unfrozen for end–to–end optimization, where the commitment and entropy losses are introduced alongside classification. The weights of these codebook–related losses are gradually ramped to stabilize convergence and balance discrete representation learning with predictive performance.

This progressive strategy preserves predictive performance in Stage 1, while reducing training instability and enabling the codebook to capture frequent substructures in Stage 2.

## 3 EXPERIMENTS

In this section, we present experiments designed to evaluate BIOBLOBS across diverse protein prediction tasks, highlighting its ability to generalize under both functional and structural challenges.

**Datasets** ProteinShake (Kucera et al., 2023)is a benchmarking toolkit that offers ready-to-use protein structure datasets, splits, and metrics across many tasks. We selected **Gene Ontology(GO)**, **EC (Enzyme Commission)**, and **Structural Class(SCOP)** to evaluate our method. In the GO dataset, we predict only the *Molecular Function* branch, where each label denotes a molecular-level activity a protein can perform (e.g., binding or catalysis), making this a multi-label task. For the EC dataset, the label is a top-level EC class that groups enzymes by reaction chemistry, such as oxidoreductases, forming a multi-class task. For the SCOP dataset, we predict *SCOP-FA*, where labels capture families defined by shared three-dimensional fold and evolutionary relatedness, again as a multi-class task focused on fold/family recognition. Across all three datasets, we evaluate under both a random split and a structure split. The structure split clusters proteins by structural similarity and assigns clusters to different splits to reduce leakage, and we use a 0.7 similarity threshold to separate closely related structures, following ProteinShake experiments. Dataset statistics are displayed in Table 1.

**Baselines and Implementations** We benchmark **GNN**, a naïve **GVP-GNN**, and **GVP+DiffPool** against two stages of BIOBLOBS. GIN follows the ProteinShake baseline on residue graphs with a task-matched classifier. The naïve GVP-GNN shares the same GVP encoder as BIOBLOBS but

Table 1: Details on Benchmark Datasets

| Dataset | # Proteins | Area of protein biology | Metric |
|---|---|---|---|
| Gene Ontology (GO) | 32,633 | Functions, Components, Pathways | $F_{max}$ |
| Enzyme Commission (EC) | 15,603 | Reaction catalysis | Accuracy |
| Structural class (SCOP) | 10,066 | Geometric relationships | Accuracy |

applies graph-level pooling over residue embeddings for classification. GVP-GNN serves as a backbone baseline to show the added value of BIOBLOBS 's architecture. DiffPool provides a comparison to learned soft clustering: after the GVP encoder, a single DiffPool layer assigns residues to clusters, then one pass of cluster-level message passing and fusion with residue features yields predictions. For BIOBLOBS, we keep hyperparameters fixed across datasets and split strategies to avoid over-tuning and to highlight robustness. Specifically, we use the same protein structure encoder as GVP-GNN, and we cap the number of blobs $T = 5$ and the blob size at $S = 15$ for neural partitioner to balance training efficiency and accuracy. We report test results for Stage 1 (BIOBLOBS w/o codebook) and Stage 2 (BIOBLOBS) to quantify the contribution of each BIOBLOBS' component. The hyperparameter settings for BIOBLOBS are reported in Appendix B.1.

Table 2: Comparison of models trained with different representations of protein structure across various tasks, evaluated on both **random** and **structure-based** data splits. Shown are mean and standard deviation across four runs with different seeds.

| Representation | Split | SCOP-FA | Enzyme Class | Gene Ontology |
|---|---|---|---|---|
| GNN | Random | $0.495 \pm 0.012$ | $0.790 \pm 0.007$ | $0.704 \pm 0.001$ |
| | Structure | $0.415 \pm 0.015$ | $0.621 \pm 0.026$ | $0.474 \pm 0.014$ |
| GVP-GNN | Random | $0.4663 \pm 0.001$ | $0.801 \pm 0.004$ | $0.569 \pm 0.032$ |
| | Structure | $0.464 \pm 0.027$ | $0.451 \pm 0.003$ | $0.504 \pm 0.004$ |
| GVP+DiffPool | Random | $0.312 \pm 0.013$ | $0.807 \pm 0.005$ | $0.583 \pm 0.001$ |
| | Structure | $0.264 \pm 0.022$ | $0.657 \pm 0.004$ | $0.458 \pm 0.016$ |
| BIOBLOBS w/o Codebook | Random | $0.495 \pm 0.009$ | $0.823 \pm 0.013$ | $0.669 \pm 0.001$ |
| | Structure | $0.442 \pm 0.001$ | $0.671 \pm 0.012$ | $0.525 \pm 0.001$ |
| BIOBLOBS | Random | $\mathbf{0.506 \pm 0.002}$ | $\mathbf{0.840 \pm 0.001}$ | $\mathbf{0.817 \pm 0.001}$ |
| | Structure | $\mathbf{0.467 \pm 0.007}$ | $\mathbf{0.684 \pm 0.004}$ | $\mathbf{0.528 \pm 0.003}$ |

## 3.1 PROTEIN CLASSIFICATION PERFORMANCE

According to Table 2, BIOBLOBS is best on all three datasets under both *random* and *structure* splits, with larger margins on the structure split that limits similarity leakage. On EC (structure), the score rises from $0.621$ for GNN to $0.684$ with BIOBLOBS (**+10%**) and from $0.451$ for GVP-GNN to $0.684$ (**+52%**). Notably, BIOBLOBS without the codebook surpasses GVP-GNN on EC and GO, indicating that the neural partitioner and the global–cluster attention fusion help the model target function-relevant local substructures instead of averaging signals over the full graph. Adding the codebook then gives further gains without hurting classification despite quantization: a VQ-VAE–style straight-through path with EMA updates and a commitment term steers blob embeddings toward a compact, reusable set of frequent, function-related patterns; on GO (random), the score increases from $0.669$ to $0.817$ (**+22%**).

GVP-GNN does not reliably outperform GNN. For the EC's structure split, it drops to $0.451$ versus $0.621$ for GNN, likely because global pooling dilutes signals from small active sites. Integrating DiffPool with GVP-GNN does not consistently improve over the baselines because one-shot soft coarsening can merge residues that are spatially distant or functionally unrelated into the same cluster. This blending weakens the functional signal of true substructures and introduces noise in the form of irrelevant or misleading features that distract the classifier. BIOBLOBS avoids these issues by growing compact local blobs with the neural partitioner and fusing them with a light attention gate, which reduces noise from disconnected parts and sharpens label alignment.

## 3.2 BioBlobs Partition Interpretation

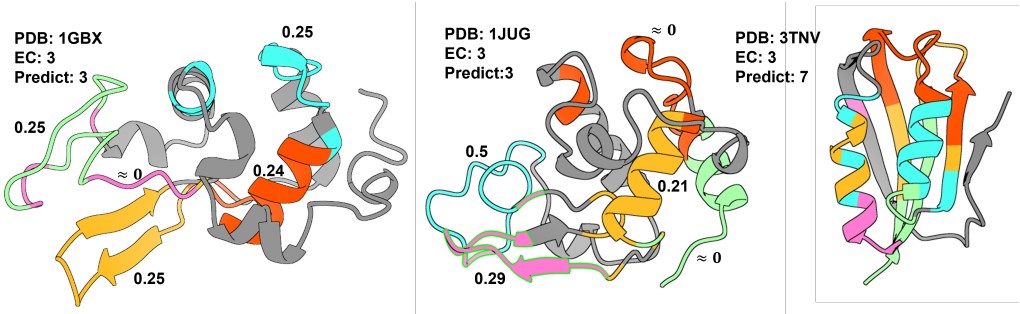

Figure 2: Actual BioBlobs partitions on the EC test set. Blobs are highlighted in distinct colors, with unassigned residues shown in gray. We mark the PDB ID, the true first level of EC number, and the predicted number by the protein. The numbers next to each blob denote its attention-based importance score $\pi_t$, discussed in Sec. 2.5

To interpret the partitions produced by BioBlobs, we constructed visualizations for proteins in the Enzyme Commission (EC) test set. For each protein, we saved the residue composition of each blob, rendered the blobs in ChimeraX (Goddard et al., 2018) for inspection, and annotated the figure with the PDB identifier, the ground-truth first-level EC number, and the model's predicted label. We begin with two correctly predicted proteins, 1GBX and 1JUG. In both cases, structurally similar substructures are partitioned consistently, revealing potential function-related motifs. The blobs highlight well-defined secondary structural elements: $\alpha$-helices (1GBX's red blob, 1JUG's yellow blob) and $\beta$-sheets (1GBX's yellow and 1JUG's pink blobs). Such patterns are biologically meaningful because many enzyme functions rely on recurring substructures. For example, the Rossmann fold—common in many dehydrogenases—combines helices and sheets to bind cofactors such as NAD or NADP (Kamiński et al., 2021).

The global-blob attention scores strengthen the interpretability of the partition. Blobs that cover function-relevant substructures tend to receive high importance values. For instance, in 1GBX, the yellow region covering a coherent beta sheet is assigned a high score. Moreover, both 1GBX and 1JUG assign nontrivial attention to flexible or connecting regions, suggesting that the model also values domain context beyond rigid motifs. In contrast, the pink region in 1GBX is disconnected and disordered, and it receives a near-zero score. This suggests that BioBlobs does not assign residues randomly, but rather emphasizes structurally stable and functionally relevant regions.

We also examine a misclassified example, 3TNV. Unlike the first two, its blobs are less connected, and individual secondary structure elements are fragmented across multiple blobs. No blob cleanly captures a coherent structural motif, making the partitioning less interpretable. This suggests that when the blob partition quality degrades, the interpretability of blob-level importance also degrades, which may contribute to errors in the final prediction. Therefore, BioBlobs offers an interpretable bridge between model predictions and protein substructure.

## 3.3 Neural Partitioner Case Study

In Fig. 3, we study how the neural partitioner parameters influence the test performance of BioBlobs on the EC and SCOP-FA datasets. We focus on two factors: the maximum blob size $S$ and the maximum number of blobs $T$, while keeping other hyperparameters fixed. For maximum blob size, we fix $T = 5$. The left figure shows that increasing $S$ from 5 to 15 gives slight performance improvements, but pushing it further to 25 causes a clear drop. This suggests that both datasets benefit from substructures of moderate size (5–10 residues), whereas overly large blobs tend to merge unrelated residues, thereby diluting function-specific signals and reducing accuracy. For the number of blobs, we fix $S = 15$. The right figure shows a similar pattern across datasets. Accuracy peaks when only five blobs are used, then declines as more blobs are introduced, likely because additional, function-irrelevant blobs compete for attention and confuse the classifier. Interestingly, when $T = 25$, the performance rebounds. At this point, nearly all residues are assigned

to blobs, which raises the chance of covering function-relevant substructures and improves prediction. Overall, both "how many" and "how large" the blobs are should accommodate the specific protein predictive task. When labels depend on short, localized motifs (for example, catalytic or binding patches), a maximum blob size of 10–15 is effective. When labels depend on domain-level organization, a larger cap around 25 may help.

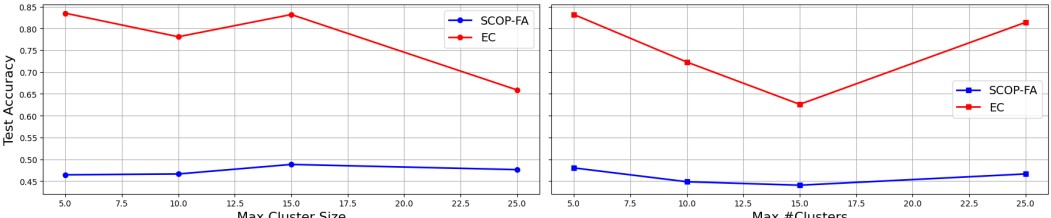

Figure 3: Neural partitioner case study: tuning maximum cluster size $S$ and number of clusters $T$.

## 4 RELATED WORK

**Protein Structure Modeling** Protein GNNs typically build residue-level graphs (distance or $k$-NN) and pass messages; equivariant backbones such as GVP remain strong. Higher-order lifts via simplicial/cell-complex networks capture multi-body interactions (Bodnar et al., 2021; Goh et al., 2022; Wang et al., 2025), and surface-centric models learn directly on molecular surfaces from MaSIF onward (Gainza et al., 2020). We also adopt residue graphs, but go beyond fixed neighborhoods with learned, connected substructures that are quantized via a codebook to better match modular organization.

**Protein Motifs and Substructures** Protein function is often mediated by recurrent modules such as catalytic triads, Rossmann cores, and P-loop NTPase sites (Dodson & Wlodawer, 1998; Rossmann et al., 1974; Leipe et al., 2003). Domain-scale taxonomies (SCOPe, CATH) catalog such units and are standard for supervision and leakage control (Chandonia et al., 2022; Orengo et al., 1997). Our approach targets finer, task-adaptive, connected "blobs" and quantizes them into a reusable substructure vocabulary.

**Graph Pooling, Subgraphs, and Partitioning** Graph pooling coarsens graphs via soft assignments/selections (DiffPool, MinCutPool, Graph U-Nets/Top-K, SAGPool, ASAP) (Liu et al., 2022), while subgraph GNNs rely on rigid extraction (Alsentzer et al., 2020). In contrast, *partitioning* assigns each node to exactly one cluster under constraints; many formulations are NP-hard (Feder et al., 1999), motivating learned relaxations and neural combinatorial methods (Karalias & Loukas, 2020; Bouritsas et al., 2021). Our layer is a differentiable partitioner for proteins: it selects seeds with straight-through Gumbel sampling and expands connected blobs within bounded neighborhoods to yield non-overlapping, size-controlled substructures.

**Discrete Representation Learning** Discrete latents turn continuous features into tokens: VQ-VAE with commitment/ST losses and its variants (VQ-VAE-2, VQGAN) connect to product/residual quantization (Van Den Oord et al., 2017; Razavi et al., 2019). Protein-specific discretization (e.g., Foldseek's alphabet) shows the value of tokenized 3D patterns (Van Kempen et al., 2024). We quantize learned blob embeddings with EMA/commitment/entropy regularization, yielding an interpretable substructure lexicon fused via global–blob attention.

## 5 CONCLUSION

BIOBLOBS is a deep protein-structure representation framework that dynamically captures the modular organization of proteins. We address the challenge of selecting substructures of variable size and topology by introducing a differentiable seed–and–expand procedure for connected node sets, coupled with discrete codebook (vector-quantized) learning. Across three protein-property prediction benchmarks, BIOBLOBS combined with GVP-based feature extractors consistently outperforms strong baselines. Together, these results lay a foundation for more faithful protein representations and a scalable, systematic account of structure–function relationships.

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

# A    APPENDIX

## USE OF LARGE LANGUAGE MODELS (LLMS)

LLMs were used to assist in coding, writing, and producing figures. All LLM-produced code and text was thoroughly double-checked.

# B    REPRODUCIBILITY

All code, data, and weights necessary to reproduce results and use our models on new data are available on `https://anonymous.4open.science/r/BioBlobs-EECD/`. Benchmarking was done with the ProteinShake Kucera et al. (2023) open source library to ensure reproducibility.

## B.1    MODEL HYPERPARAMETERS

**Default Hyperparameters.**    We summarize the default hyperparameters used in the two–stage training of BIOBLOBS across 3 datasets and 2 splits.

**Stage 1 (baseline training).**    We train the encoder, partitioner, and global–cluster attention fusion for 120 epochs with batch size 128 and learning rate $10^{-3}$. The GVP encoder uses scalar–vector input dimensions of $(6, 3)$ for residues and $(32, 1)$ for edges, with hidden dimensions $(100, 16)$ and $(32, 1)$, respectively. We stack three GVP convolution layers with dropout rate 0.1 and apply sum pooling for graph readout. The neural partitioner is configured with up to $T = 5$ clusters, hidden dimension 50, $k = 1$ hop expansion, maximum cluster size 15, termination threshold 0.95, and temperature annealing from $\tau_{\text{init}} = 1.0$ to $\tau_{\text{min}} = 0.1$ with exponential decay 0.95.

**Stage 2 (joint training with codebook).**    We resume from the Stage 1 checkpoint and initialize the substructure codebook using $K$–means clustering on the pre-trained cluster embeddings. The codebook has 256 entries of dimension 100 (matching the scalar hidden dimension) and is updated by EMA with decay 0.99. Training continues for 30 epochs with reduced learning rate $10^{-4}$ and without backbone freezing. Additional objectives are introduced: the commitment loss with weight $\lambda_{\text{vq}} = 1.0$ and the entropy regularizer with $\lambda_{\text{ent}} = 0.1$ This staged procedure allows the encoder and partitioner to first stabilize on meaningful blobs, after which the codebook is optimized jointly without destabilizing the backbone.

## B.2 Neural Blob Partitioner Algorithm

---

**Algorithm 1** Neural BLOB Partitioner

---

**Require:** Residue embeddings $\mathbf{Z} \in \mathbb{R}^{N \times D}$, edge index $\mathcal{E}$, max blobs $T$, max blob size $S$, $k$-hop radius $k$, temperature schedule $\{\tau^{(t)}\}$, termination threshold $\rho$, step size $\alpha_g$

**Ensure:** Assignment $\mathbf{M} \in \{0, 1\}^{N \times T}$, blob embeddings $\mathbf{H} = \left[\mathbf{h}_{\text{clus}}^{(1)}, \ldots, \mathbf{h}_{\text{clus}}^{(T)}\right]$

1: $\mathbf{g}^{(0)} \leftarrow \text{Pool}(\mathbf{Z})$; $\mathbf{M} \leftarrow \mathbf{0}$; $\mathbf{m} \leftarrow \mathbf{1}_N$            $\triangleright$ global context, assignment, availability

2: **for** $t = 1$ to $T$ **do**

3:      **Early stop:** if $\text{coverage}(\mathbf{M}) \geq \rho$ or no $i$ with $m_i = 1$, **break**

4:      **Seed selection**

5:      $\boldsymbol{\ell}^{(t)} \leftarrow f_{\text{seed}}\left([\mathbf{Z}; \mathbf{g}^{(t-1)}]\right)$; mask unavailable: $\tilde{\boldsymbol{\ell}}^{(t)} = \boldsymbol{\ell}^{(t)} + \log \mathbf{m}$

6:      $\mathbf{w}^{(t)} \leftarrow \text{GumbelSoftmax}\left(\tilde{\boldsymbol{\ell}}^{(t)}, \tau^{(t)}\right)$          $\triangleright$ hard variant with straight-through (ST)

7:      $\mathbf{e}^{(t)} \leftarrow \text{one\_hot}(\arg\max_i w_i^{(t)})$; $\tilde{\mathbf{w}}^{(t)} \leftarrow \mathbf{e}^{(t)} + (\mathbf{w}^{(t)} - \text{sg}(\mathbf{w}^{(t)}))$

8:      $\mathbf{z}_{\text{seed}}^{\text{ST}} \leftarrow \sum_i \tilde{w}_i^{(t)} \mathbf{z}_i$; $i_s \leftarrow \arg\max_i e_i^{(t)}$

9:      $\mathbf{M}[i_s, t] \leftarrow 1$; $m_{i_s} \leftarrow 0$

10:     **Candidate set and local stats**

11:     $\mathcal{R}^{(t)} \leftarrow \mathcal{N}_k(i_s; \mathcal{E}) \cap \{i : m_i = 1\}$               $\triangleright$ $k$-hop and available

12:     $\boldsymbol{\phi}^{(t)} \leftarrow \text{LocalStats}(\mathcal{R}^{(t)})$       $\triangleright$ log#cands, seed-link fraction, induced density

13:     **Threshold prediction**

14:     $\theta^{(t)} \leftarrow f_{\text{thresh}}\left([\mathbf{z}_{\text{seed}}^{\text{ST}}; \mathbf{g}^{(t-1)}; \boldsymbol{\phi}^{(t)}]\right)$

15:     **Blob expansion**

16:     For $i \in \mathcal{R}^{(t)}$: $s_i^{(t)} = \frac{(\mathbf{W}_k \mathbf{z}_i)^\top (\mathbf{W}_q \mathbf{z}_{\text{seed}}^{\text{ST}})}{\sqrt{d}}$      $\triangleright$ optionally add normalized link term

17:     $p_i^{(t)} = \sigma\left((s_i^{(t)} - \theta^{(t)})/\tau^{(t)}\right)$; $c_i^{(t)} = \mathbb{I}[p_i^{(t)} \geq 0.5]$         $\triangleright$ ST hard gate

18:     Keep the seed and the top-$S$ residues by $p_i^{(t)}$; set other $c_i^{(t)} = 0$

19:     For all $i$: $\mathbf{M}[i, t] \leftarrow c_i^{(t)}$; $m_i \leftarrow m_i \cdot (1 - c_i^{(t)})$

20:     **Blob embedding (ST) and context update**

21:     $\mathbf{h}_{\text{hard}}^{(t)} = \frac{\sum_i c_i^{(t)} \mathbf{z}_i}{\sum_i c_i^{(t)} + \varepsilon}$,    $\mathbf{h}_{\text{soft}}^{(t)} = \frac{\sum_i \tilde{p}_i^{(t)} \mathbf{z}_i}{\sum_i \tilde{p}_i^{(t)} + \varepsilon}$,    $\mathbf{h}_{\text{clus}}^{(t)} = \mathbf{h}_{\text{hard}}^{(t)} + \left(\mathbf{h}_{\text{soft}}^{(t)} - \text{sg}(\mathbf{h}_{\text{soft}}^{(t)})\right)$

22:     $\mathbf{g}^{(t)} \leftarrow \mathbf{g}^{(t-1)} + \alpha_g \, f_{\text{global}}(\mathbf{h}_{\text{clus}}^{(t)})$

23: **end for**

24: **return** $\mathbf{M}$, $\mathbf{H}$

---

## B.3 EMA UPDATE OF THE CODEBOOK

We maintain a vector-quantization codebook $\mathbf{E} = \{\mathbf{e}_k\}_{k=1}^{K}$ and update it with exponential moving averages (EMA) as in VQ-VAE (Van Den Oord et al., 2017). Let $\mathcal{H} = \{\mathbf{h}_s\}$ be the set of cluster embeddings in a minibatch, and let $r_{s,k} \in \{0, 1\}$ denote the hard assignment of $\mathbf{h}_s$ to code $k$ (or soft responsibilities in a soft variant). Define the batch counts and sums

$$n_k = \sum_s r_{s,k}, \qquad \mathbf{u}_k = \sum_s r_{s,k}\, \mathbf{h}_s.$$

With decay $\lambda \in (0, 1)$, EMA keeps running totals $\hat{N}_k$ and $\hat{\mathbf{U}}_k$:

$$\hat{N}_k^{(t)} = \lambda\, \hat{N}_k^{(t-1)} + (1 - \lambda)\, n_k, \qquad \hat{\mathbf{U}}_k^{(t)} = \lambda\, \hat{\mathbf{U}}_k^{(t-1)} + (1 - \lambda)\, \mathbf{u}_k,$$

and updates each code vector to the normalized running mean

$$\mathbf{e}_k^{(t)} = \frac{\hat{\mathbf{U}}_k^{(t)}}{\hat{N}_k^{(t)} + \varepsilon}.$$

We apply stop-gradient to $\hat{N}_k^{(t)}$ and $\hat{\mathbf{U}}_k^{(t)}$ so that code updates do not backpropagate into the encoder. If the accumulators start at zero, an optional bias correction divides numerator and denominator by $(1 - \lambda^t)$. A small $\varepsilon$ prevents division by zero and stabilizes the update when counts are low. In practice, $\lambda \in [0.99, 0.999]$ works well; codes with near-zero effective count can be reinitialized to recent cluster embeddings to avoid dead entries. The EMA path replaces direct gradient descent on $\mathbf{E}$ and tracks the empirical means of assigned embeddings with lower variance when assignments fluctuate.

