# OpenReview forum: "BioBlobs: Differentiable Graph Partitioning for Protein Representation Learning"
_ICLR.cc/2026/Conference — Submitted to ICLR 2026_

### Official Review · Reviewer_Utcj · 2025-10-27

**Soundness:** 3
**Presentation:** 2
**Contribution:** 2
**Rating:** 4
**Confidence:** 4

**Summary:**

This paper presents BIOBLOBS, a fully differentiable module for adaptive, structure-aware graph partitioning in protein representation learning (PRL). Instead of using fixed local neighborhoods (e.g., k-hop or voxel grids), BIOBLOBS dynamically partitions protein graphs into cohesive, non-overlapping substructures—termed blobs—whose size and topology are optimized end-to-end. The resulting blob embeddings are quantized via a learnable codebook, forming a discrete vocabulary of functionally meaningful substructures. The authors integrate BIOBLOBS atop GVP-GNN encoders and demonstrate improved performance on multiple PRL benchmarks (e.g., protein function and binding prediction), along with enhanced interpretability through blob importance maps.

**Strengths:**

1. The paper introduces a differentiable graph-partitioning mechanism that adapts substructure granularity during training. This approach bridges neural combinatorial optimization and PRL, representing a conceptual advance over rigid, hand-defined neighborhoods. The use of Gumbel–Softmax partitioning and codebook quantization to learn discrete, interpretable substructures is a creative fusion of ideas from clustering, representation learning, and structural biology.

2. Methodologically, the paper is sound. The authors provide a detailed formulation of the partitioning process (seed selection, expansion, masking, quantization), demonstrate end-to-end differentiability, and support claims with ablations on blob size, number of partitions, and codebook usage.

3. The experiments compare BIOBLOBS to strong baselines such as GVP-GNN and show consistent gains across several datasets. Controls for structural leakage and granularity analysis indicate careful experimental design.

**Weaknesses:**

1. The experiments focus on a limited set of functional prediction benchmarks. It would strengthen the case to evaluate BIOBLOBS on more diverse downstream tasks (e.g., structure-based docking, mutational effect prediction, or protein–ligand binding affinity), demonstrating its generality.

2. While baselines like GVP-GNN are strong, the paper does not include comparisons to recent hierarchical or modular protein encoders (e.g., UniFold-H, StructLM, or modular-GNN variants). This omission makes it difficult to quantify improvements over existing multi-scale architectures.

3. The differentiable partitioning and codebook quantization likely increase computational cost. The paper does not quantify the added complexity or runtime relative to standard PRL models—important for adoption in large-scale protein modeling.

4. Although interpretability via blob importance is discussed, biological case studies (e.g., mapping discovered blobs to catalytic motifs or known domains) are missing. Without such validation, interpretability claims remain qualitative.

5. The ablations on blob number and size are informative but could be extended to explore sensitivity to temperature scheduling, codebook dimensionality, or the impact of quantization loss on downstream accuracy.

**Questions:**

1. Does the learned partitioning remain stable across runs or vary stochastically? Reporting variance or overlap between runs would clarify the determinism of the discovered substructures.

2. The discrete quantization step is conceptually interesting—could the authors analyze how many unique blob types are learned and whether they correspond to recurring structural motifs (e.g., β-α-β loops, Rossmann folds)?

3. How does the model behave on very large proteins (>1000 residues)? Are memory or efficiency issues encountered, and could hierarchical or approximate partitioning help?

4. Could BIOBLOBS operate in latent space (e.g., over embeddings from ESM-3) to discover “latent substructures” when 3D coordinates are unavailable?

5. The framework could identify de novo substructures linked to function. Have the authors analyzed any newly discovered blobs for potential biological significance?

---

### Official Review · Reviewer_Ezq6 · 2025-10-27

**Soundness:** 2
**Presentation:** 3
**Contribution:** 3
**Rating:** 4
**Confidence:** 4

**Summary:**

This paper proposes BioBlobs, a differentiable graph-partitioning framework for protein representation learning. The method introduces a neural partitioner that segments protein graphs into coherent substructures (“blobs”), followed by a vector-quantized codebook to discretize substructure embeddings, and a global–blob attention module for aggregation. The model is evaluated on the ProteinShake benchmark and compared with GIN, GVP-GNN, and GVP+DiffPool.

**Strengths:**

1. The idea of introducing a differentiable and context-adaptive partitioning module to model protein substructures is technically novel and interesting.

2. The integration of vector quantization (VQ-VAE style) to construct a reusable substructure vocabulary is innovative and well-motivated for interpretability.

3. Visual analyses show promising qualitative interpretability by highlighting functionally coherent regions.

4. The method is clearly presented with reproducibility support. The comprehensive ablation studies have verified the rationality of proposed components.

**Weaknesses:**

1. Motivation not compelling. The necessity of explicit structural partitioning in protein representation learning is not well justified. Strong geometric learning models such as ProNet [1] and GearNet [2] already achieve high performance without such partitioning. The claimed benefit over standard geometric encoders remains unclear beyond architectural novelty.

2. Limited evaluation scope. All experiments rely solely on the ProteinShake benchmark. This does not convincingly demonstrate generalization, especially given the benchmark’s narrow task diversity. Evaluation on additional datasets (e.g., Fold classification, enzyme active-site prediction) would make the results more persuasive.

3. Insufficient baselines. Only GIN, GVP-GNN (w/o DiffPool) are used for comparison. Missing recent geometric and equivariant protein models weakens the empirical support. I understand that some baselines may not work on ProteinShake dataset, so incorporating another dataset may also works for this point.

4. No runtime or efficiency evidence. Section 2.6 presents complexity analysis, but does not report actual computational time compared with a baseline GVP model . As the encoder remains GVP-based, this method clearly introduces non-trivial additional cost due to iterative seed selection, expansion, and quantization, yet the overhead is not quantified. Without runtime profiling, efficiency claims are unsupported.

[1] Wang, Limei, et al. "Learning hierarchical protein representations via complete 3d graph networks." arXiv preprint arXiv:2207.12600 (2022).

[2] Zhang, Zuobai, et al. "Protein representation learning by geometric structure pretraining." arXiv preprint arXiv:2203.06125 (2022).

**Questions:**

The idea is interesting, and the architectural design demonstrates clear technical creativity. The experimental results, particularly the interpretability analysis, are insightful and indicate potential for future development. However, the paper would benefit from more rigorous empirical validation, especially regarding runtime efficiency and comparison with stronger baselines.

I am willing to modify my score after further revision and discussion.

---

### Official Review · Reviewer_MFGX · 2025-11-01

**Soundness:** 3
**Presentation:** 2
**Contribution:** 2
**Rating:** 4
**Confidence:** 4

**Summary:**

The authors introduced a differentiable graph partitioning method for protein structure representation learning. The method uses a quantized codebook and aggregate blob embedding with attention mechanism. This method aims to map substructure onto a discrete latent code book on top of a GVP encoder. While individual components are preciously studied in this context, the combination of them is novel.  Limited results shows improvement of performance on tasks such as GO/EC number prediction and SCOP-FA benchmarks over pure GVP encoder.

**Strengths:**

The main strength of the paper is it introduced a novel sub-structuring method for structure based protein encoding which improves the based model interpretability and sees gain in performance.

1. The authors presented a novel substructure pooling/partitioning method with graph partitioning and applied it on top of a GVP structure encoder to install model interpretability in the GVP based structure encoder.
2. The modular nature of the model allows this to be applied to various structure based encoder for interpretability.
3. The adaptive segmentation allows the model to learn task specific and structurally relevant features in contrast to distance based graph encoding.
4. More computationally efficient compared to DiffPool.

**Weaknesses:**

The main weakness of the paper are in-depth analysis of the interpretability claim and limited benchmarking.

1. The final attention fusion mechanism seems under motivated, can a simpler module achieve comparable performance?
2. There are many other structure encoders out there but only GVP is presented.
3. Lacking comparison with non-differentiable hard partitioners such as https://arxiv.org/pdf/1511.02074 and HGP-SL(https://arxiv.org/abs/1911.05954).

**Questions:**

1. For the neural partitioner, the seed is sampled separately for each protein. In inference time will the seed affect the final segmentation significantly? For example, if a protein is sampled with different seeds across runs, how will the model behave? There is not experiment showcasing this.
2. For the Blob expansion, the illustration indicates the segmentation follows certain intrinsic protein substructure like secondary structure motifs, is this the case for the actual partitioner?
3. Does the codebook size matter int these type of model?
4. How does this method scale to larger protein? Does the number of the blobs matter in those context?

---

### Official Review · Reviewer_hPga · 2025-11-01

**Soundness:** 3
**Presentation:** 3
**Contribution:** 2
**Rating:** 4
**Confidence:** 4

**Summary:**

This paper introduces BIOBLOBS, a fully differentiable, plug-and-play module for protein representation learning (PRL) that addresses a critical limitation of existing models: their reliance on rigid substructures (e.g., k-hop neighborhoods, fixed voxels) which distort functional signals from flexible, modular protein substructures.
BIOBLOBS frames protein "atomization" (breaking proteins into meaningful units) as a graph-partitioning task. Its pipeline includes four core components: (1) a protein graph encoder (using Geometric Vector Perceptrons, GVP) to extract residue-level features; (2) a neural blob partitioner that iteratively selects seed residues (via Gumbel-Softmax) and expands them into non-overlapping, size-controlled "blobs" (cohesive substructures); (3) a substructure codebook that quantizes blobs into a discrete, reusable vocabulary of function-relevant patterns (via vector quantization with EMA updates); and (4) a global-blob attention fusion module that integrates quantized blobs with global protein context to produce final embeddings.

**Strengths:**

1. A novel, task-adaptive partitioner that guarantees blob connectivity, avoids overlapping substructures, and integrates with existing encoders (e.g., GVP-GNN). Systematic ablations of maximum blob size (S) and number (T) reveal that moderate-sized blobs (5–15 residues) and 5–10 blobs per protein optimize performance, aligning with biological modularity.
2. By linking blobs to known motifs (e.g., Rossmann folds), it enables researchers to interpret why a model makes a prediction—closing the gap between black-box deep learning and biological understanding.

**Weaknesses:**

1. The current design uses fixed T=5 (max blobs) and S=15 (max blob size), which may fail to capture interactions between large domains (e.g., proteins with 500+ residues or 3+ domains). The time complexity (O(T(k_hop k_NN +1)N)) also raises concerns for massive datasets (e.g., AlphaFold DB with 200M+ proteins).
2. The codebook is claimed to capture "function-relevant substructures," but there is no quantitative validation of which biological motifs (e.g., catalytic triads, P-loops) the codebook entries represent. It is unclear if entries are enriched for known functional patterns or random noise.
3. The global-blob attention fusion is critical to weighting blob importance. It is unknown how much attention contributes to performance.
4. As lfor The EC and GO tasks, the proposed method are not compard with other prevalent methods, like GearNet, SaProt, etc.

**Questions:**

How does BIOBLOBS handle proteins with highly disordered regions (e.g., >30% disordered residues)?

---

### Meta-Review · Area_Chair_jaQV · 2026-01-06

**Summary:**

All reviewers are consistently concerned about this paper and the authors do not provide any rebuttals.

**Reviewer Concerns:**

The authors do not provide any rebuttals.

**Reviewer Scores:**

All reviewers are consistently concerned about this paper with low scores.

---

### Decision · Program_Chairs · 2026-01-26

Reject